# AUTOREGRESSIVE IMAGE GENERATION WITH RANDOMIZED PARALLEL DECODING

**Haopeng Li**[1][*]  **Jinyue Yang**[2]  **Guoqi Li**[2][†]  **Huan Wang**[1][†]

[1]Westlake University  [2]Institute of Automation, Chinese Academy of Sciences

https://github.com/hp-l33/ARPG

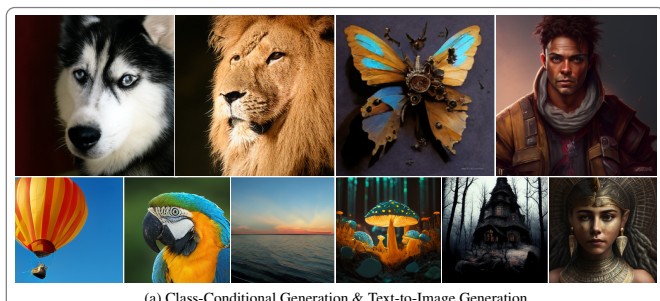
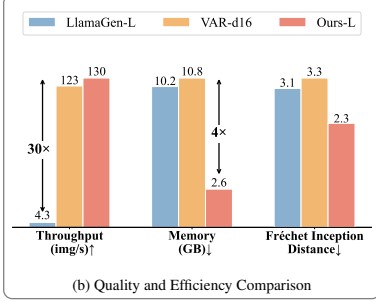

(a) Class-Conditional Generation & Text-to-Image Generation

(b) Quality and Efficiency Comparison

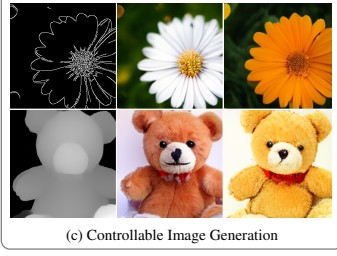
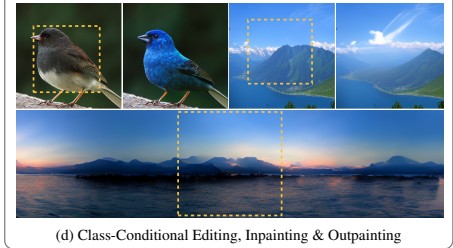
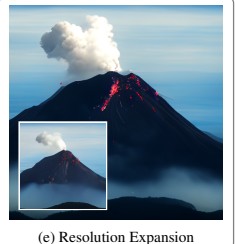

(c) Controllable Image Generation

(d) Class-Conditional Editing, Inpainting & Outpainting

(e) Resolution Expansion

Figure 1: **Visual Autoregressive Model with Randomized Parallel Generation (ARPG):** A high-quality and efficient framework for image synthesis. ARPG enables (a) class-conditional and text-to-image generation with just 64-step parallel decoding and (b) outperforms recent representative works (*e.g.*, VAR (Tian et al., 2024), LlamaGen (Sun et al., 2024a)) in throughput, memory consumption, and quality. It further supports (c) controllable generation and zero-shot generalization, including (d) class-conditional editing, inpainting, outpainting, and (e) resolution expansion.

## ABSTRACT

We introduce **ARPG**, a novel visual **A**utoregressive model that enables **R**andomized **P**arallel **G**eneration, addressing the inherent limitations of conventional raster-order approaches, which hinder inference efficiency and zero-shot generalization due to their sequential, predefined token generation order. Our key insight is that effective random-order modeling necessitates explicit guidance for determining the position of the next predicted token. To this end, we propose a novel *decoupled decoding* framework that decouples positional guidance from content representation, encoding them separately as queries and key-value pairs. By directly incorporating this guidance into the causal attention mechanism, our approach enables fully random-order training and generation, eliminating the need for bidirectional attention. Consequently, ARPG readily generalizes to *zero-shot inference* tasks such as image inpainting, outpainting, and resolution expansion. Furthermore, it supports *parallel inference* by concurrently processing multiple queries using a shared KV cache. On the ImageNet-1K 256×256 benchmark, our approach attains an FID of 1.83 with only 32 steps, achieving a **30×** and **3×** speedup in inference over raster-order and recent parallel AR models, respectively, while reducing memory consumption by **75%** at a similar scale.

---

[*]Work done during the author's research internship.

[†]Corresponding authors: <wanghuan@westlake.edu.cn>, <guoqi.li@ia.ac.cn>.

# 1 INTRODUCTION

Autoregressive (AR) models have demonstrated remarkable performance and scalability (Henighan et al., 2020; Kaplan et al., 2020), particularly in the domain of large language models, where the next-token prediction paradigm has driven significant advancements (Achiam et al., 2023; et al., 2024). This success has extended to visual synthesis, enabling breakthroughs in autoregressive image generation with methods like VQGAN (Esser et al., 2021; Ramesh et al., 2021; Lee et al., 2022). However, directly applying next-token prediction to images presents fundamental challenges. Unlike text, which possesses an inherent causal structure, images are defined over a two-dimensional spatial domain. AR models necessitate flattening this spatial information into a sequence, typically following a rigid, predefined order (e.g., raster-scan). This strictly sequential generation process is not only inefficient, especially for high-resolution images, but also fundamentally limits the model's ability to perform zero-shot generalization to tasks requiring non-causal dependencies.

To address these challenges, alternative approaches, such as MaskGIT (Chang et al., 2022), have adopted a masked modeling (Devlin et al., 2019) approach for parallel token generation in random order. However, the reliance on bidirectional attention prevents the use of the KV cache, resulting in high computational overhead. Block-wise AR (Tian et al., 2024; Liu et al., 2024; Wang et al., 2025; Chang et al., 2023) enable block-wise parallel decoding, however, they are constrained by fixed block orderings and sampling schedules, which limit their flexibility and fidelity. Recent work RandAR (Pang et al., 2025) enables fully random-order training and inference with causal attention via positional instruction tokens, but incurs significant memory and computational costs due to the increased sequence length. Therefore, a fundamental challenge persists: *how to achieve flexible-order and parallel generation without compromising computational efficiency*.

**In this work**, we introduce *ARPG*, a novel visual AR model that enables training and inference in fully random token orders through a *decoupled decoding* process. Unlike the standard decoder-only Transformers (Vaswani et al., 2017), our approach decouples the prediction process into two distinct passes: (1) The *content refinement pass* utilizes causal self-attention over a random-order token sequence to construct label-leakage-free content representations as key-value pairs, without directly predicting tokens. (2) In the *position-guided prediction pass*, data-independent [MASK] tokens endowed with positional information corresponding to a right-shift of the input, act as target-aware queries. These queries use causal cross-attention to predict their respective target tokens based on the content key-value pairs processed in the first pass. This design allows the model to be trained within a fully causal paradigm while enabling generalization to block-wise parallel decoding with flexible token orders, as multiple queries can be processed independently in a single step.

Extensive experiments demonstrate the superiority of ARPG. On the ImageNet-1K 256×256 benchmark (Deng et al., 2009), ARPG at various scales achieve FID (Heusel et al., 2017) of 2.30, 1.93, and 1.83. This performance surpasses recent works while achieving a nearly 30× and 3× speedup over raster-order and parallel AR models, respectively, and reducing memory usage by 75%.

Furthermore, we extend our method to more complex tasks, including controllable generation, zero-shot generalization, and text-to-image generation, as shown in Fig. 1. In summary:

1. We propose a novel visual autoregressive framework that enables parallel image generation with a random token order using a decoupled two-pass decoding mechanism, overcoming the inefficiencies and poor generalization of traditional next-token prediction methods.
2. We explore the zero-shot generalization ability of our method and further extend it to versatile controllable generation and text-to-image generation.
3. Extensive experiments demonstrate that our approach achieves competitive generation quality while simultaneously excelling in throughput and memory consumption, setting a new benchmark for efficient, high-performance autoregressive image generation.

# 2 RELATED WORK

## 2.1 AUTOREGRESSIVE IMAGE GENERATION

State-of-the-art large language models (Achiam et al., 2023; et al., 2024) adopt a decoder-only Transformer (Vaswani et al., 2017) for causal modeling of language sequences and autoregressive gener-

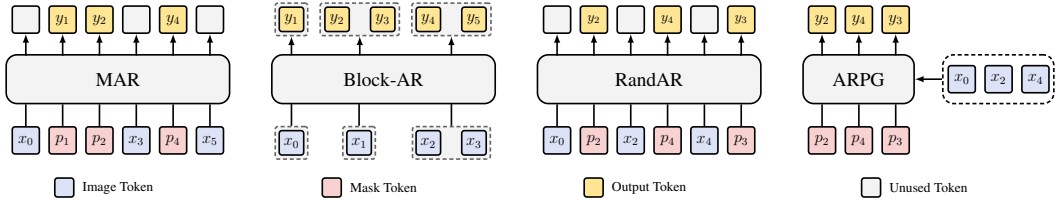

Figure 2: **Methods for representing the position of the next token.** **MAR** (Chang et al., 2022; Li et al., 2024b) indicates the position via masking out image token; **Block-AR** models (Tian et al., 2024; Wang et al., 2025; He et al., 2025) use predefined positions; **RandAR** (Pang et al., 2025) intersperses position tokens throughout the sequence; and our **ARPG** integrates it as a query in a cross-attention mechanism.

ation, a method commonly known as the GPT (Achiam et al., 2023) style approach. In the vision domain, images can be quantized into discrete tokens (Van Den Oord et al., 2017) and flattened from 2D to 1D, enabling generation via the next-token prediction paradigm, as seen in models like VQGAN (Esser et al., 2021), LlamaGen (Sun et al., 2024a), etc. (Lee et al., 2022; Wang et al., 2024; Li et al., 2024a; Wang et al., 2024; Yu et al., 2024). These methods have demonstrated impressive generative performance. However, this token-by-token image generation approach is inefficient, especially when dealing with high-resolution images. Additionally, since the generation can only proceed in a specific token order, it encounters difficulties in zero-shot inference that require non-causal dependencies, such as inpainting and outpainting.

## 2.2 Acceleration of Visual Autoregressive Models

Another mainstream approach to sequence modeling is the encoder-only architecture. This method, widely used in language models like BERT (Devlin et al., 2019), involves randomly masking and then predicting multiple tokens in a sequence. In the vision domain, this paradigm is adopted by models such as MaskGIT (Chang et al., 2022) for image generation. By leveraging bidirectional attention, these masked-generation methods eliminate causal dependencies, enabling multi-token generation in a single step and thus significantly faster inference. However, due to the absence of a KV cache, their overall inference efficiency remains limited. Beyond this, other parallel generation techniques face their own challenges. Many block-wise autoregressive methods (Tian et al., 2024; Wang et al., 2025; He et al., 2025) are constrained by predefined orders and schedulers, while various training-free methods (Teng et al., 2025) achieve acceleration at the cost of degraded output quality. A recent work, RandAR (Pang et al., 2025), inspired by XLNet (Yang et al., 2019), implements a permuted autoregressive model for parallel decoding by interspersing position tokens throughout the sequence. This strategy, however, incurs a heavy penalty: it doubles the sequence length, thereby increasing both the computational load and the memory required for the KV cache.

## 3 Method

### 3.1 Rethinking Visual Autoregressive Modeling

**Preliminaries.** Given a sequence $\mathbf{X} = \{x_1, x_2, \ldots, x_n\}$, a causal autoregressive model predicts the next token based on all preceding tokens, following the probabilistic formulation:

$$p(\mathbf{x}) = \prod_{i=1}^{n} p(x_i \mid x_1, x_2, \ldots, x_{i-1}). \tag{1}$$

The formulation for block-wise AR models (Tian et al., 2024; Wang et al., 2025) is analogous, substituting a single token $x_i$ with a set of tokens $\{x_i, x_{i+1}, ..., x_{i+j}\}$.

In contrast, masked sequence modeling (Devlin et al., 2019) processes sequences with certain tokens masked out and predicts them based on the unmasked context. The objective is to minimize the negative log-likelihood of the tokens at masked positions (Devlin et al., 2019):

$$\mathcal{L} = -\mathbb{E}\Big[ \sum_{m_i=1} \log p(x_i \mid \mathbf{X_M})\Big], \quad \forall i \in \{1, 2, \cdots, n\}, \tag{2}$$

where $\mathbf{M} = \{m_1, m_2, \ldots, m_n\} \in \{0, 1\}^n$ is masked positions and $\mathbf{X_M}$ is the corrupted sequence.

Figure 3: **Analysis of attention scores.** Normalized attention maps from multiple distinct heads in the final layer of RandAR (Pang et al., 2025). The maps, partitioned by token type (masked vs. unmasked), reveal that attention weights are predominantly concentrated on unmasked tokens.

**Key Insights.** Based on the characteristics of the methods discussed, we find:

*Insight 1: Breaking the order-specific constraints of AR model requires explicit positional guidance.*

This insight stems from the fundamental difference in how models determine the prediction target. As shown in Fig. 2, causal autoregressive models are bound to a strict, predefined generation order (e.g., raster-scan for standard AR, scale or diagonal patterns for block-wise AR), which inherently limits their flexibility. Masked modeling using position-aware [MASK] tokens to act as explicit instructions, directing the model to the specific locations that need to be filled. This mechanism of providing direct positional guidance is what allows the model to predict tokens in a flexible manner.

*Insight 2: Masked modeling is inherently training-inefficient due to its sparse prediction objective.*

This inefficiency arises because the loss function is calculated exclusively on [MASK] tokens, which represent only a fraction of the input sequence. More precisely, the query vectors corresponding to unmasked tokens receive no *direct* gradients from the optimization objective. Consequently, in any given training step, the model parameters are suboptimally updated using only a subset of the tokens, yet this process incurs the full computational cost equivalent to processing the entire sequence. The related proof is provided in Appendix C.

*Insight 3: Attention directed towards [MASK] tokens is redundant.*

This redundancy is empirically evident, as exemplified by the RandAR (Pang et al., 2025) attention scores in Fig. 3, which show that [MASK] tokens contribute minimally to the attention mechanism. The vast majority of attention scores are allocated to unmasked tokens. We argue this is also conceptually sound based on the roles of different tokens. An unmasked token should attend to other unmasked tokens to enrich its semantic representation. A [MASK] token, tasked with reconstructing the original token at its position, should attend to semantic-rich unmasked tokens to gather contextual information. In neither case is it necessary to attend to other [MASK] tokens, which contain no native semantic content.

## 3.2 VISUAL AUTOREGRESSIVE MODEL WITH RANDOMIZED PARALLEL DECODING

**Reformulation.** We observe that the essential components for predicting a token $x_{\tau_t}$ at position $\tau_t$ within an arbitrary permutation $\mathcal{T}$ of indices are: (1) the set of already known tokens $\{x_{\tau_i}\}_{i=1}^{t-1}$ and (2) the target position $\tau_t$ itself. The state of other unknown positions is irrelevant. This motivates a reformulation where the prediction is explicitly conditioned only on these necessary elements:

$$\prod_{t=1}^{n} p\big(x_{\tau_t} \mid x_{\tau_1}, x_{\tau_2}, \ldots, x_{\tau_{t-1}}\big) = f_\theta\big(\{x_{\tau_i}\}_{i=1}^{t-1}, \ \tau_t\big). \tag{3}$$

Here, $f_\theta$ is a model parameterized by $\theta$ that takes the known tokens and the target position as separate inputs. Based on *Insights 2* and *3*, we configure the softmax attention (Vaswani et al., 2017) such that queries are derived exclusively from data-independent [MASK] tokens, while the keys and values originate entirely from unmasked content tokens. This decoupling yields three key advantages. **(1)** The content representations can be learned independently. **(2)** Decoding is guided by positional queries that attend to rich, fully-learned KV representations, rather than the less informative ones from shallow layers. **(3)** The projection parameters of query are not shared with key-value, enabling full learning in a single training step without redundancy computation. These points constitute the key distinctions from prior works (Yang et al., 2019; Pang et al., 2025; Liu et al., 2024).

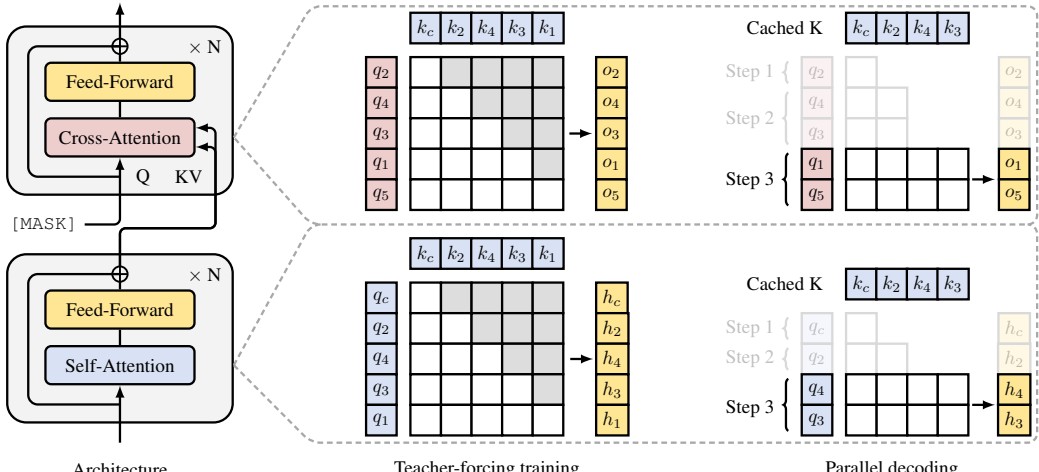

Figure 4: **Architecture**: The 1st decoder extract representations of image tokens. The 2nd decoder use target-aware [MASK] tokens as queries that attend to key-value pairs from the output of the 1st decoder. **Teacher-forcing training** is performed under a causal attention. **Parallel decoding** is achieved by inputting multiple queries in a single step, with each query independently attending to existing KV cache (omit value for clarity).

**Two-Pass Decoder Architecture.** Based on the preceding discussion, we employ a two-pass decoder architecture to decouple the prediction of a target token $x_{\tau_t}$ from the representation learning of known tokens $\{x_{\tau_i}\}_{i=1}^{t-1}$. The equivalence of the probabilistic model is maintained through the application of a suitable causal mask. As illustrated in Fig. 4 and formulated as:

1. **Pass-1: Content Representation Learning.** The first decoder consists of a standard causal self-attention. Unlike typical autoregressive models that predict the next token, its sole purpose is to process the sequence of known content token embeddings $\{x_{\tau_i}\}_{i=1}^{t-1}$ to generate a set of rich, context-aware representations $\{h_{\tau_i}\}_{i=1}^{t-1}$. These representations are then projected into key-value pairs, which serve as a comprehensive summary of all historical information.

2. **Pass-2: Position-Guided Decoding.** The second decoder is comprised of causal cross-attention. In this pass, query vector $q_{\tau_t}$ are derived from the [MASK] token, and infused with positional information for a specific target location. This target-aware query attends to key-value pairs that are derived from $\{h_{\tau_i}\}_{i=1}^{t-1}$ produced by Pass-1, ultimately predicting predict $\hat{x}_{\tau_t}$.

Mathematically, let $m \in \mathbb{R}^{1 \times d}$ be the embedding of [MASK] token, for a given target position $\tau_t$ and content tokens represented by $\{h_{\tau_i}\}_{i=1}^{t-1}$ from Pass-1. Using the rotary position embedding (RoPE) (Su et al., 2024), the attention operations in layer $l$ of Pass-2 are:

$$q_{\tau_t}^{(l)} = \text{RoPE}(o_{\tau_t}^{(l-1)} W_q^{(l)}, \tau_t), \quad \forall \tau_t, o_{\tau_t}^{(0)} = m. \tag{4}$$

$$k_{\tau_i}^{(l)} = \text{RoPE}(h_{\tau_i} W_k^{(l)}, \tau_i), \quad v_{\tau_i}^{(l)} = h_{\tau_i} W_v^{(l)}. \tag{5}$$

$$o_t^{(l)} = q_{\tau_t}^{(l)} + \text{Attention}(q_{\tau_t}^{(l)}, \{k_{\tau_j}^{(l)}\}_{j=1}^{t-1}, \{v_{\tau_j}^{(l)}\}_{j=1}^{t-1}), \tag{6}$$

where $W_q, W_k, W_v \in \mathbb{R}^{d \times d}$ are learnable projection matrices.

**Training and Decoding.** **(1)** During training, the Pass-1 decoder learns causal representations from a shuffled sequence. Then the positional information of the sequence is right-shifted by one position and embedded into [MASK] tokens to serve as target-aware queries, as illustrated in Fig. 4. **(2)** At inference time, we first compute the KV cache from the known tokens using the self-attention in Pass-1. Next, we select multiple target-aware queries. These queries can simultaneously attend to the KV cache via cross-attention in Pass-2, thereby enabling multi-token prediction within a single inference step, as illustrated in Fig. 4. In this setting, the causal attention at step $t$ in Pass-1 can be *generalized to block-wise attention* so that the model learns local bidirectional representations for the tokens generated in step $t-1$. This mismatch in attention patterns does not compromise the causal conditional probability model. Instead, it improves image fidelity (validated in Sec.4) and increases robustness to diverse sampling schedules, including the cosine and arccos schemes used in methods such as MaskGiT (Chang et al., 2022) and MUSE (Chang et al., 2023).

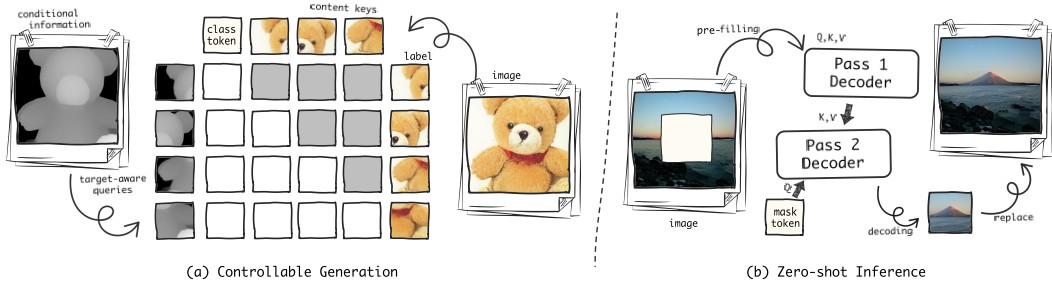

Figure 5: **Implementation details.** (a) Conditional inputs provide the queries. (b) For zero-shot inpainting, known regions are pre-filled in Pass-1, while masked regions are generated in Pass-2.

**Controllable Generation & Zero-shot Inference.** (1) Controllable generation is analogous to class-conditional generation, but it uses representations from conditional inputs (e.g., depth maps) as queries instead of positional [MASK] tokens. As illustrated in Fig. 5. (2) Zero-shot inference is achieved via a two-stage process. In the case of inpainting (Fig. 5), the Pass-1 decoder is first pre-filled with tokens from the known image regions. Subsequently, the target regions are replaced by [MASK] tokens, and the Pass-2 decoder generates them using the key-value pairs from the Pass-1. This process is also applicable to outpainting and resolution expansion.

**Comparison with Other Methods.** In contrast to recent studies and their notable limitations, ARPG is fundamentally distinct. The key differences are detailed below and summarized in Tab. 1.

Different block-wise AR methods introduce unique limitations. VAR (Tian et al., 2024) relies on a multi-scale image tokenizer for coarse-to-fine prediction, a strategy that significantly increases the token count and computational overhead. PAR (Wang et al., 2025), SAR (Liu et al., 2024), and NAR (He et al., 2025) predict parallel spatial blocks but are constrained by a rigid, predefined ordering or scheduler that impairs sample quality and scheduling flexibility.

Table 1: **Summary of existing methods.**

| Method | Attention Pattern | Scheduler | Zero-shot |
|---|---|---|---|
| MAR | Bidirectional | Cosine | ✓ |
| VAR | Block-wise | Predefined | ✓ |
| PAR | Block-wise | Predefined | ✗ |
| NAR | Block-wise | Predefined | ✗ |
| RAR | Causal | N/A | ✗ |
| RandAR | Causal | Flexible | ✓ |
| ARPG | Causal → Block-wise | Flexible | ✓ |

RAR (Yu et al., 2024) randomly permutes sequences during training but progressively anneals them to a raster-scan order, meaning *it does not support random-order parallel generation in practice*.

RandAR (Pang et al., 2025) couples the position and content tokens. This method not only doubles the computational cost but also leads to suboptimal fidelity, as the less-informative content representations in shallow layers are forced to co-learn with [MASK] token. Furthermore, it necessitates extra memory to cache [MASK] tokens and retains a causal attention during parallel decoding.

# 4 EXPERIMENTS

## 4.1 IMPLEMENTATION DETAILS

**Model.** We implement three models of different scales, all of which follow the LlamaGen design (Sun et al., 2024a) and use its tokenizer. All layers in the Pass-2 decoder share a global key-value projection to improve efficiency (Sun et al., 2024b). More details shown in Appendix A.

**Training.** We train the class-conditional model on ImageNet-1K (256×256) (Deng et al., 2009) for 400 epochs and the text-to-image model on a 4M subset of BLIP-3o (Chen et al., 2025) for 50 epochs. All models are optimized with the AdamW (Loshchilov & Hutter, 2019) optimizer using a learning rate of 1e-4 per 256 batch size. Further configuration details are in Appendix A.

**Metrics.** We use FID (Heusel et al., 2017) as the primary evaluation metric, and additionally report Inception Score (IS) (Salimans et al., 2016), precision, and recall (Kynkäänniemi et al., 2019), following the protocol of ADM (Dhariwal & Nichol, 2021). For text-to-image generation, we evaluate our models using selected metrics from GenEval (Ghosh et al., 2023). The efficiency profile is evaluated on an NVIDIA A800-80GB GPU, considering only the token generation process.

Table 2: **Overall comparisons on ImageNet benchmarks**. Arrows indicate whether lower or higher is better. Efficiency was profiled with a batch size of 64 and bfloat16 precision.

| Type | Model | Param. | Steps | Throughput (img/s)↑ | Mem. (GB)↓ | FID↓ | IS↑ | Pre.↑ | Rec.↑ |
|------|-------|--------|-------|---------------------|------------|------|-----|-------|-------|
| *Resolution: 256×256* | | | | | | | | | |
| Diffusion | DiT-L/2 (Peebles & Xie, 2023) | 458 M | 250 | 1.32 | 1.62 | 5.02 | 167.2 | 0.75 | 0.57 |
| | DiT-XL/2 (Peebles & Xie, 2023) | 675 M | 250 | 0.91 | 2.14 | 2.27 | 278.2 | 0.83 | 0.57 |
| Mask | MaskGIT (Chang et al., 2022) | 227 M | 8 | 46.18 | 1.71 | 6.18 | 182.1 | 0.80 | 0.51 |
| | MAR-B (Li et al., 2024b) | 208 M | 100 | 1.71 | 1.47 | 2.31 | 281.7 | 0.82 | 0.57 |
| | MAR-L (Li et al., 2024b) | 479 M | 100 | 1.27 | 2.32 | 1.78 | 296.0 | 0.81 | 0.60 |
| VAR | VAR-d16 (Tian et al., 2024) | 310 M | 10 | 123.21 | 10.85 | 3.30 | 274.4 | 0.84 | 0.51 |
| | VAR-d20 (Tian et al., 2024) | 600 M | 10 | 75.38 | 15.97 | 2.57 | 302.6 | 0.83 | 0.56 |
| | VAR-d24 (Tian et al., 2024) | 1.0 B | 10 | 49.94 | 22.29 | 2.09 | 312.9 | 0.82 | 0.59 |
| AR | LlamaGen-L (Sun et al., 2024a) | 343 M | 576 | 4.33 | 10.23 | 3.07 | 256.1 | 0.83 | 0.52 |
| | LlamaGen-XL (Sun et al., 2024a) | 775 M | 576 | 2.46 | 17.11 | 2.62 | 244.1 | 0.80 | 0.57 |
| | LlamaGen-XXL (Sun et al., 2024a) | 1.4 B | 576 | 1.58 | 26.22 | 2.62 | 244.1 | 0.80 | 0.57 |
| | RAR-B (Yu et al., 2024) | 261 M | 256 | 14.12 | 4.65 | 1.95 | 290.5 | 0.82 | 0.58 |
| | RAR-L (Yu et al., 2024) | 461 M | 256 | 12.08 | 6.37 | 1.70 | 299.5 | 0.81 | 0.60 |
| | RAR-XL (Yu et al., 2024) | 955 M | 256 | 8.00 | 10.55 | 1.50 | 306.9 | 0.80 | 0.62 |
| AR (Parallel) | PAR-L (Wang et al., 2025) | 343 M | 147 | 14.77 | 10.25 | 3.76 | 218.9 | 0.81 | 0.60 |
| | PAR-XL (Wang et al., 2025) | 775 M | 147 | 7.91 | 17.13 | 2.61 | 259.2 | 0.80 | 0.62 |
| | PAR-XXL (Wang et al., 2025) | 1.4 B | 147 | 5.23 | 26.25 | 2.35 | 263.2 | 0.80 | 0.62 |
| | NAR-L (He et al., 2025) | 372 M | 31 | 42.71 | 10.25 | 3.06 | 263.9 | 0.81 | 0.53 |
| | NAR-XL (He et al., 2025) | 816 M | 31 | 23.97 | 17.13 | 2.70 | 277.5 | 0.81 | 0.58 |
| | NAR-XXL (He et al., 2025) | 1.5 B | 31 | 15.23 | 26.25 | 2.58 | 293.5 | 0.82 | 0.57 |
| | RandAR-L (Pang et al., 2025) | 343 M | 88 | 25.30 | 7.32 | 2.55 | 288.8 | 0.81 | 0.58 |
| | RandAR-XL (Pang et al., 2025) | 775 M | 88 | 15.51 | 13.52 | 2.25 | 317.8 | 0.80 | 0.60 |
| | RandAR-XXL (Pang et al., 2025) | 1.4 B | 88 | 10.46 | 21.77 | 2.15 | 322.0 | 0.79 | 0.62 |
| AR (Parallel) | ARPG-L | 320 M | 32 | 130.14 | 2.78 | 2.30 | 297.7 | 0.82 | 0.56 |
| | ARPG-XL | 719 M | 32 | 80.56 | 4.65 | 1.93 | 349.2 | 0.80 | 0.61 |
| | ARPG-XXL | 1.3 B | 32 | 55.28 | 7.22 | 1.83 | 336.1 | 0.80 | 0.60 |
| | ARPG-L | 320 M | 64 | 67.47 | 2.64 | 2.37 | 293.7 | 0.82 | 0.55 |
| | ARPG-XL | 719 M | 64 | 45.09 | 4.57 | 1.99 | 340.6 | 0.80 | 0.61 |
| | ARPG-XXL | 1.3 B | 64 | 31.65 | 7.18 | 1.86 | 339.7 | 0.81 | 0.59 |
| *Resolution: 512×512* | | | | | | | | | |
| | DiT-XL/2 (Peebles & Xie, 2023) | 675 M | 250 | 0.18 | 4.70 | 3.04 | 240.8 | 0.84 | 0.54 |
| | MaskGIT (Chang et al., 2022) | 227 M | 12 | 4.48 | 7.63 | 7.32 | 156.0 | 0.78 | 0.50 |
| | VQGAN (Esser et al., 2021) | 1.4 B | 1024 | 0.63 | 44.12 | 26.52 | 66.8 | 0.73 | 0.31 |
| | VAR-d36 (Tian et al., 2024) | 2.0 B | 10 | - | OOM | 2.63 | 303.2 | - | - |
| | ARPG-XL | 719 M | 64 | 35.53 | 13.98 | 2.82 | 277.5 | 0.82 | 0.56 |

(a) **Class-conditional generation.**      (b) **Controllable generation.**

Figure 6: **Generation samples.** ARPG can efficiently generate high-fidelity images with 64 steps

## 4.2 IMAGE GENERATION

**Class-Conditial Generation.** We compare the results of ARPG with existing methods on the ImageNet-1K, as shown in Tab. 2. For 256×256 benchmarks, we present two sampling settings: 32 steps for optimal FID and 64 steps for the best visual quality. Samples are shown in Fig. 6a.

Compared to LlamaGen (Sun et al., 2024a), ARPG-XXL achieves a better FID of 1.83 with only 32 sampling steps, representing a **30× speedup** with superior fidelity. Compared to VAR (Tian et al., 2024), ARPG achieves higher throughput while **reducing 75% memory** consumption. Furthermore, compared to recent parallel decoding methods (Wang et al., 2025; He et al., 2025; Pang et al., 2025), ARPG achieves better generation quality with superior efficiency.

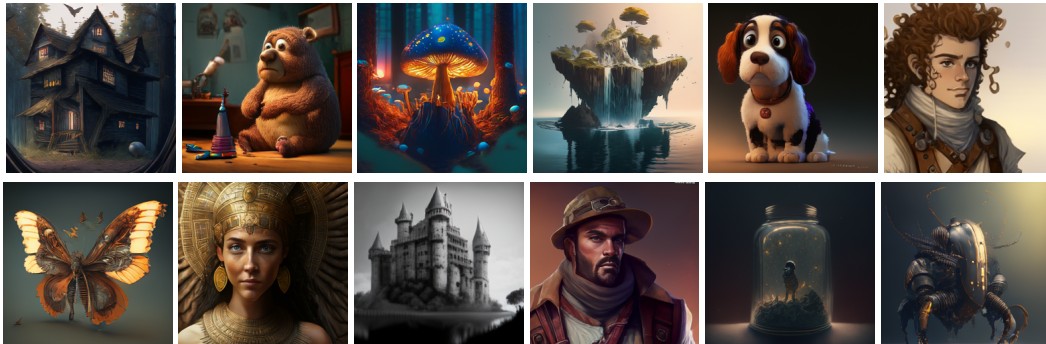

Figure 7: **Text-to-image generation.** ARPG-XL can efficiently generate high-quality images with only 64 steps based on the given image captions. Prompt omitted for brevity.

Table 3: **Quantitative evaluation of text-to-image generation at 512×512 resolution.**

| Model | Params. | Data | Single Obj. | Two Obj. | Colors | Overall | Throughput |
|---|---|---|---|---|---|---|---|
| LlamaGen-XL (Sun et al., 2024a) | 0.8 B | 60 M | 0.87 | 0.34 | 0.64 | 0.32 | 0.83 img/s |
| Chameleon (Team, 2024) | 7.0 B | 1.4 B | - | - | - | 0.39 | - |
| SD-v1.5 (Rombach et al., 2022) | 0.9 B | 2 B | 0.97 | 0.38 | 0.76 | 0.43 | 4.32 img/s |
| ARPG-XL | 0.8 B | 4 M | 0.87 | 0.30 | 0.68 | 0.31 | 30.11 img/s |

For 512×512 resolution, we fine-tune the XL model (pre-trained at 256×256 resolution) instead of training from scratch to save computational resources. With only 50 epoch fine-tuning, it achieves competitive quality while surpassing others in throughput, as shown in Tab. 2.

**Text-to-Image Generation.** We use the pre-trained FLAN-T5-XL (Chung et al., 2024) to generate prompt embeddings that condition the token sequence. As shown in Tab. 3, our method achieves performance comparable to LlamaGen (Sun et al., 2024a) using only **7% of the training data** while offering significantly higher throughput. Generated samples are shown in Fig. 7.

**Controllable Generation.** Following ControlAR (Li et al., 2024d), we use a pre-trained ViT (Dosovitskiy et al., 2021) adapter to process conditional inputs, such as Canny edges and depth maps. These conditional inputs are used as target-aware queries to fine-tune the pre-trained class-conditional model. As shown in Tab. 4 and Fig. 6b, ARPG significantly outperforms recent works such as ControlVAR (Li et al., 2024c) and ControlAR (Li et al., 2024d).

Table 4: **Controllable generation on ImageNet.**

| Method | Model | Param. | FID↓ Canny | FID↓ Depth |
|---|---|---|---|---|
| ControlVAR | VAR-d16 | 310M | 16.20 | 13.80 |
| | VAR-d20 | 600M | 13.00 | 13.40 |
| | VAR-d30 | 2.0B | 7.85 | 6.50 |
| ControlAR | AiM-L | 350M | 9.66 | 7.39 |
| | LlamaGen-L | 343M | 7.69 | 4.19 |
| ControlARPG | ARPG-L | 320M | 7.39 | 4.06 |

**Zero-shot Generalization.** We assess the zero-shot generalization capabilities of ARPG on a variety of image manipulation tasks, with qualitative results presented in Fig. 8. It demonstrates that ARPG excels at inpainting (class-conditional editing) and outpainting without any task-specific fine-tuning, generating content that is both high-fidelity and semantically consistent with the surrounding context. This strong performance highlights a key advantage of our flexible, random-order generation. More zero-shot inference samples of different tasks are provided in the Appendix D.

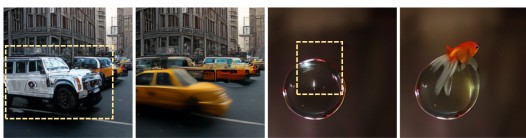

(a) **Left:** image inpainting. **Right:** image editing.

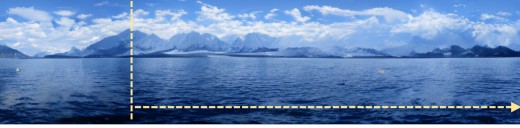

(b) **Outpainting:** from 256×256 to 1024×256

Figure 8: **Zero-shot inference of different tasks.**

Table 5: **Ablation study of model design.** The baseline model is ARPG-L, trained for 150 epochs with 64 sampling steps. "Rand. & Parall." denotes support for randomized parallel generation.

| Description | Parameters | Layers | Rand. & Parall. | Steps | Throughput↑ | Memory↓ | FID↓ | IS ↑ |
|---|---|---|---|---|---|---|---|---|
| ARPG-L | 320 M | 12+12 | ✓ | 64 | 67.47 img/s | 2.64 GB | 3.51 | **282.7** |
| + w. CosinePE | 320 M | 12+12 | ✓ | 64 | 70.63 img/s | 2.64 GB | 3.61 | 262.4 |
| + w/o Shared KV | 343 M | 12+12 | ✓ | 64 | 48.02 img/s | 3.83 GB | **3.46** | 228.1 |
| Fewer Pass-2 Decoder | 332 M | 18+ 6 | ✓ | 64 | 62.35 img/s | 3.34 GB | 3.82 | 223.0 |
| More Pass-2 Decoder | 307 M | 6+18 | ✓ | 64 | 71.24 img/s | 1.93 GB | 3.51 | 242.5 |
| Fully Pass-2 Decoder | 295 M | 0+24 | ✓ | 64 | 72.26 img/s | 0.91 GB | 4.57 | 255.9 |
| w/o. Pass-2 Decoder | 343 M | 24+ 0 | ✗ | 256 | 11.70 img/s | 4.96 GB | >90 | <50 |

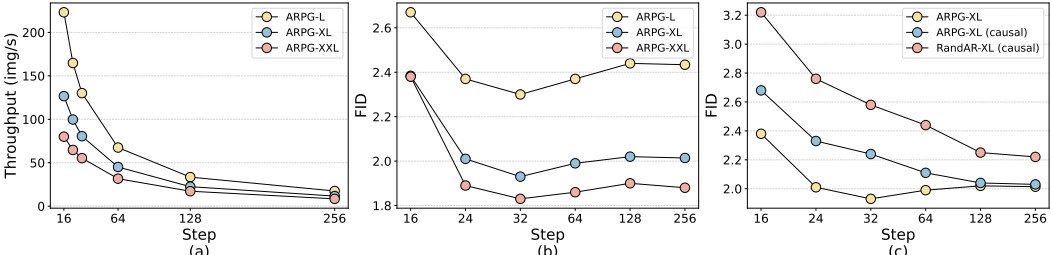

Figure 9: **Ablation study of parallel decoding.** The impact of decoding steps on speed (a) and quality (b). Generalized attention patterns enhance quality with fewer decoding steps (c).

## 4.3 ABLATION STUDY

We report several key ablation studies. Additional experiments are presented in the Appendix B.

**Effect of Decoding Steps.** We evaluated how the number of decoding steps affects both generation quality and efficiency. As shown in Fig. 9, reducing the number of sampling steps substantially improves inference efficiency while not significantly degrading generation quality. In some cases, quantitative quality metrics at fewer steps even outperform those obtained with more steps.

**Effect of Attention Pattern.** We additionally evaluated a setting in which the attention pattern is prevented from generalizing from causal to block-wise attention at inference. The results in Fig. 9 show the opposite trend to the permissive setting, namely that fewer steps produce markedly worse quality. These findings support our claim in Sec. 3 that generalizing causal attention to block-wise attention does not invalidate the underlying probabilistic model. On the contrary, the resulting local bidirectional awareness helps the model better predict future tokens.

**Effect of Architecture Design.** We explored the impact of architecture design, as shown in Tab. 5.

1. **Decoder.** The higher the proportion of the Pass-2 decoders, the higher the inference efficiency, but the more severe the deterioration of the generation quality. Reducing the proportion of the Pass-2 decoder not only reduces inference efficiency but also degrades the generation quality. When there are no Pass-2 decoders at all, the model degenerates into a standard AR model and loses the ability of randomized parallel decoding.

2. **KV Projection.** We also examined the impact of using shared KV, as discussed in the Sec. 3. Without shared KV, although the generation quality of the model is slightly improved, it significantly affects the inference speed and memory consumption. To balance the generation quality and inference efficiency, we choose to use the shared KV design in subsequent experiments.

3. **Position Encoding.** We further examine the effect of different positional encodings. Replacing RoPE with cosine positional encodings leads to a degradation in performance.

## 5 CONCLUSION

In this work, we propose a novel autoregressive image generation framework that can parallelly generate images in random token orders, breaking the limitations of the inefficiency of the next token prediction paradigm and its poor zero-shot generalization ability.

ACKNOWLEDGMENTS

This work is supported by the Young Scientists Fund of the National Natural Science Foundation of China (NSFC) (No. 62506305), the Zhejiang Leading Innovative and Entrepreneur Team Introduction Program (No. 2024R01007), the Key Research and Development Program of Zhejiang Province (No. 2025C01026), the Scientific Research Project of Westlake University (No. WU2025WF003), and the Chinese Association for Artificial Intelligence (CAAI) & Ant Group Research Fund - AGI Track (No. 2025CAAI-ANT-13). It is also supported by the research funds of the National Talent Program and the Hangzhou Municipal Talent Program.

Additionally, this work is partially supported by the CAS Project for Young Scientists in Basic Research (YSBR-116), the National Natural Science Foundation of China (NSFC) (No. 62325603, No. 62236009, No. U22A20103), the Beijing Science and Technology Plan (No. Z241100004224011), and Shanghai NeuHelium Neuromorphic Technology Co., Ltd.

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

## A  IMPLEMENTATION DETAILS

Table 6: **ARPG model architecture and hyperparameters configuration**. $X+Y$ layers: $X$ layers in the first-pass decoder and $Y$ layers in the second-pass decoder.

| | Table 2 ARPG-L $256 \times 256$ | Table 2 ARPG-XL $256 \times 256$ | Table 2 ARPG-XXL $256 \times 256$ | Table 2 ARPG-XL $512 \times 512$ | Table 4 ARPG-L $256 \times 256$ |
|---|---|---|---|---|---|
| **Architecture** | | | | | |
| Layer | 12 + 12 | 18 + 18 | 24 + 24 | 18 + 18 | 12 + 12 |
| Hidden Size | 1024 | 1280 | 1536 | 1280 | 1024 |
| Heads | 16 | 20 | 24 | 20 | 16 |
| Parameters | 320 M | 719 M | 1.3 B | 719 M | 320 M |
| **Optimization** | | | | | |
| Training Iteration | 500 K | 670 K | 1 M | 500 K | 60 K |
| Batch Size | 1024 | 768 | 512 | 128 | 1024 |
| Optimizer | AdamW | AdamW | AdamW | AdamW | AdamW |
| - lr | 4e-4 | 3e-4 | 2e-4 | 5e-5 | 4e-4 |
| - lr scheduler | Cosine | Cosine | Cosine | Linear | Constant |
| - warmup ratio | 0.25 | 0.25 | 0.25 | 1.0 | 0.0 |
| - weight decay | 0.05 | 0.05 | 0.05 | 0.05 | 0.05 |
| - $(\beta_1, \beta_2)$ | (0.9,0.95) | (0.9,0.95) | (0.9,0.95) | (0.9,0.95) | (0.9,0.95) |
| Data Augmentation | Ten-crop | Ten-crop | Ten-crop | Ten-crop | Ten-crop |
| **Sampling** | | | | | |
| Steps Scheduler | Arccos | Arccos | Arccos | Arccos | Arccos |
| CFG Scheduler | Linear | Linear | Linear | Linear | Linear |
| Temperature | 1.0 | 1.0 | 1.0 | 1.0 | 1.0 |
| Top-K | None | None | None | None | None |
| Top-P | 1.0 | 1.0 | 1.0 | 1.0 | 1.0 |

**Architecture Configuration.**  We show the model architecture configuration in three different scales in Tab. 6. The notation X+Y layers represents the number of layers in the first-pass and second-pass decoders, respectively. For the parameter counts of the models in the controllable generation, we only report the parameters of the base model. The parameters of the additionally introduced ViT-Adapter (Li et al., 2024d), used for processing conditional information, are excluded to maintain consistency with the representation in ControlAR (Li et al., 2024d).

**Training Hyperparameters.**  We detail the hyperparameters for the experimental setups to ensure the precise reproducibility of our findings. The AdamW (Kingma, 2014; Loshchilov & Hutter, 2019) optimizer, with a consistent configuration, was utilized in all experiments. The batch size and learning rate were tailored for each model based on computational resources.

**Sampling Hyperparameters.**  For the sampling process, all models employ an arccos schedule to determine the number of tokens to decode at each step. A linear scheduler is utilized for the classifier-free guidance (CFG) (Ho & Salimans, 2022) scale. Optimal performance is achieved by setting both the temperature and top-p to 1.0, while top-k sampling is not applied (denoted as None in the Tab. 6, which is equivalent to setting top-k to the vocabulary size of 16,384).

## B  ADDITIONAL EXPERIMENTS

**Effects of Sampling Order.**  We also evaluated the performance of ARPG under specific orders, including raster order and several alternatives (Esser et al., 2021), as shown in Tab. 7. While random-order modeling is more challenging due to the $n!$ possible orderings, it still outperforms fixed orders. The constraint of a fixed order impedes effective parallel decoding and zero-shot tasks.

Table 7: **Effect of generation orders.**

| Order | Steps | FID↓ | IS↑ | Pre.↑ | Rec.↑ |
|---|---|---|---|---|---|
| Raster | 256 | 2.49 | 277.6 | 0.79 | 0.58 |
| Spiral-in | 256 | 3.71 | 221.1 | 0.75 | 0.57 |
| Spiral-out | 256 | 4.11 | 210.5 | 0.74 | 0.56 |
| Z-curve | 256 | 2.56 | 278.2 | 0.78 | 0.51 |
| Alternate | 256 | 2.56 | 279.4 | 0.78 | 0.54 |
| Random | 64 | 2.44 | 287.1 | 0.81 | 0.55 |

**Effects of Sampling Steps.**  In Tab. 8 we present a more detailed set of experiments that quantify the effects of decoding steps and attention pattern generalization on model performance. For each decoding step configuration, we carried out a fine-grained search over the classifier-free guidance (CFG) scale to ensure that the reported FID values are optimal. FID is used as the primary quantitative metric and the Inception Score (IS) is treated as a secondary metric. Accordingly, the IS values

Table 8: **Ablation study of parallel decoding.** $w_{\text{cfg}}$: the optimal classifier-free guidance scale.

| Model | Steps ($w_{\text{cfg}}$) | FID↓ | | IS↑ | | Throughput (img/s) ↑ | Memory (GB) ↓ |
|---|---|---|---|---|---|---|---|
| | | causal | block-wise | causal | block-wise | | |
| ARPG-L | 16 (7.8) | 3.51 | 2.67 | 351.09 | 302.50 | 223.18 | 3.00 |
| | 24 (6.1) | 2.96 | 2.37 | 320.73 | 293.72 | 164.80 | 2.87 |
| | 32 (5.4) | 2.69 | 2.30 | 305.71 | 287.90 | 130.14 | 2.78 |
| | 64 (5.1) | 2.59 | 2.37 | 302.63 | 297.72 | 67.47 | 2.64 |
| | 128 (4.6) | 2.45 | 2.44 | 286.82 | 289.36 | 33.44 | 2.53 |
| | 256 (4.3) | 2.43 | 2.44 | 277.59 | 278.55 | 17.45 | 2.32 |
| ARPG-XL | 16 (8.4) | 2.68 | 2.38 | 364.46 | 331.83 | 126.66 | 4.78 |
| | 24 (7.0) | 2.33 | 2.01 | 353.89 | 339.06 | 99.74 | 4.71 |
| | 32 (6.6) | 2.24 | 1.93 | 352.62 | 342.43 | 80.56 | 4.65 |
| | 64 (5.3) | 2.11 | 1.99 | 327.57 | 323.92 | 45.09 | 4.57 |
| | 128 (5.1) | 2.04 | 2.02 | 325.86 | 322.69 | 22.43 | 4.49 |
| | 256 (5.0) | 2.03 | 2.01 | 318.77 | 318.94 | 11.60 | 4.41 |
| ARPG-XXL | 16 (9.3) | 2.49 | 2.38 | 353.07 | 326.11 | 79.97 | 7.25 |
| | 24 (8.1) | 2.18 | 1.89 | 350.20 | 341.00 | 64.70 | 7.20 |
| | 32 (6.5) | 2.04 | 1.83 | 337.26 | 328.89 | 55.28 | 7.22 |
| | 64 (6.3) | 1.94 | 1.86 | 337.76 | 334.57 | 31.65 | 7.18 |
| | 128 (6.1) | 1.93 | 1.90 | 336.41 | 336.34 | 17.20 | 7.16 |
| | 256 (6.0) | 1.90 | 1.88 | 334.96 | 337.34 | 8.41 | 7.23 |

shown in Tab. 8 correspond to the operating points that minimize FID. Finally, although causal and block-wise causal attention (in Fig. 2) patterns exhibit minor differences in inference throughput and memory footprint, we report only the block-wise resource measurements in Tab. 8.

Our results demonstrate that generalizing from causal attention to block-wise causal attention during inference does not compromise the probabilistic model, but instead significantly improves performance. This finding validates our claim in Sec. 3 that the local bidirectional context is beneficial. This marks a key distinction from previous works (Pang et al., 2025; Wang et al., 2025; He et al., 2025), which are restricted to the same attention pattern for both training and inference.

**Analysis of Scalability.** To analyze the scalability of our models, we present their training loss curves on class-conditional generation tasks in Fig. 10. A clear performance hierarchy emerges based on model scale.

The largest model (1.3B parameters) achieves the lowest final loss of approximately 7.03 after 1M iterations. The model with 719M parameters converges to a loss of 7.29, while the smallest model (320M parameters) finishes with the highest loss at 7.50.

This scaling trend is a strong indicator of the robustness of our model and suggests potential for further improvement with even larger scale.

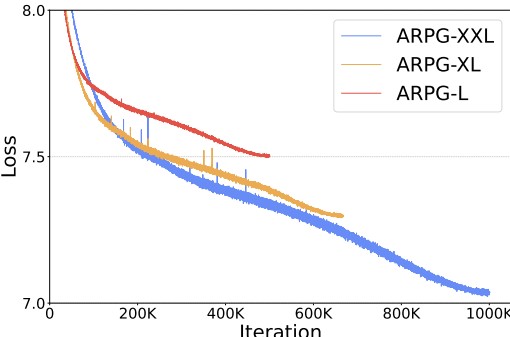

Figure 10: **Training loss curves.** All models were trained for the same epochs.

**Analysis of Attention Scores.** As we mentioned in Sec. 2, decoupling the representation learning of unmasked tokens from the prediction of [MASK] tokens addresses the computational redundancy issue. This problem arises in coupled architectures where [MASK] tokens receive negligible attention. We visualize the final-layer attention maps of ARPG's two decoders in Fig. 11. The visualization reveals that this decoupled structure leads to more uniformly distributed attention scores in both decoders. The attention patterns primarily exhibit a natural decay over long distances, avoiding the issue seen in Fig. 3 where a significant number of tokens are assigned extremely low weights.

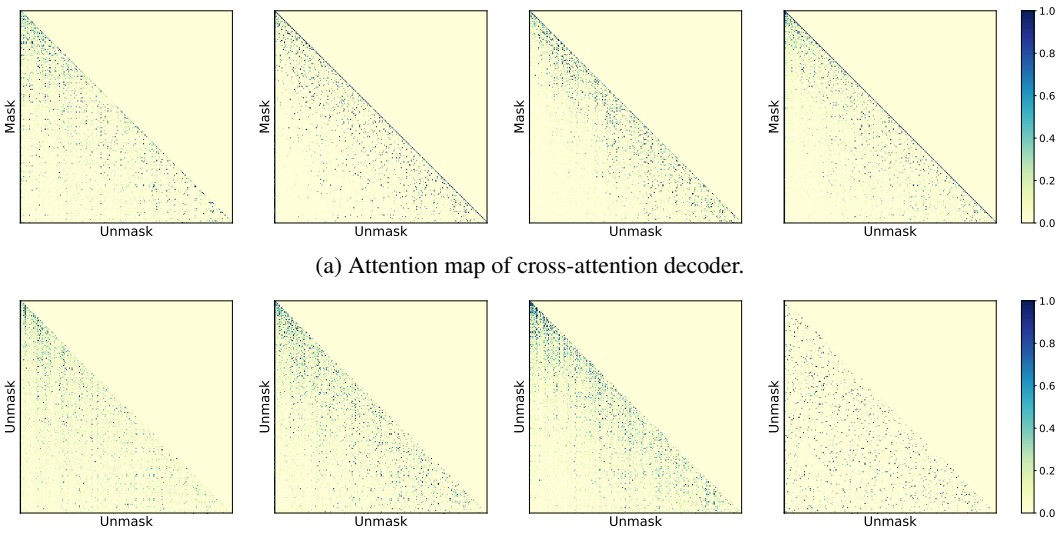

(a) Attention map of cross-attention decoder.

(b) Attention map of self-attention decoder.

Figure 11: **Attention maps of different decoders.** Visualization of normalized attention maps from different decoders. Each column corresponds to the same attention head.

## C  THEORETICAL PROOF

As mentioned in Sec. 3, we claimed that the masked modeling is inefficient due to the non-directly gradient flow caused by sparse optimization objects. Here is the theoretical proof:

*Proof.* In the final layer of the Transformer (Vaswani et al., 2017) blocks, given input sequences with length $n$, and $\boldsymbol{q}, \boldsymbol{k}, \boldsymbol{v} \in \mathbb{R}^{1 \times d}$. The softmax attention (Vaswani et al., 2017) mechanism (the scale term is omitted for clarity) operates as:

$$\boldsymbol{o}_i = \frac{\sum_{j=1}^{n} \exp(\boldsymbol{q}_i \boldsymbol{k}_j^\top) \boldsymbol{v}_j}{\sum_{j=1}^{n} \exp(\boldsymbol{q}_i \boldsymbol{k}_j^\top)} \in \mathbb{R}^{1 \times d} . \tag{7}$$

Let $\boldsymbol{S}_{ij} = \boldsymbol{q}_i \boldsymbol{k}_j^\top$, $\boldsymbol{P}_{ij} = \mathrm{softmax}(\boldsymbol{S}_i)_j$. It is evident that $d\boldsymbol{P}_{ij} = d\boldsymbol{o}_i \boldsymbol{v}_j^\top$ and the derivative of the softmax function is its Jacobian matrix. Using the fact that the Jacobian of $\boldsymbol{y} = \mathrm{softmax}(\boldsymbol{x})$ is $\mathrm{diag}(\boldsymbol{y}) - \boldsymbol{y}^\top \boldsymbol{y}$ (Dao et al., 2022), we have:

$$d\boldsymbol{S}_i = d\boldsymbol{P}_i(\mathrm{diag}(\boldsymbol{P}_i) - \boldsymbol{P}_i^\top \boldsymbol{P}_i) \tag{8}$$

$$= \boldsymbol{P}_i \odot d\boldsymbol{P}_i - (d\boldsymbol{o}_i \boldsymbol{o}_i^\top)\boldsymbol{P}_i , \tag{9}$$

$$d\boldsymbol{S}_{ij} = \boldsymbol{P}_{ij}(d\boldsymbol{P}_{ij} - d\boldsymbol{o}_i \boldsymbol{o}_i^\top) , \tag{10}$$

where $\odot$ denotes the Hadamard product. Then we derive gradient for $\boldsymbol{q}$ using chain rule:

$$d\boldsymbol{q}_i = \sum_{j=1}^{n} d\boldsymbol{S}_{ij} \boldsymbol{k}_j = \sum_{j=1}^{n} \boldsymbol{P}_{ij}(d\boldsymbol{o}_i \boldsymbol{v}_j^\top - d\boldsymbol{o}_i \boldsymbol{o}_i^\top)\boldsymbol{k}_j . \tag{11}$$

Obviously, when $\boldsymbol{o}_i$ corresponds to an unmasked token, it does not contribute to the loss calculation, resulting in $d\boldsymbol{o}_i = 0$, and consequently, $d\boldsymbol{q}_i = 0$. □

The query vectors for unmasked tokens in shallower layers are updated via indirect gradients propagated through key-value pairs from deeper layers, which provide a partial learning signal not directly tied to the optimization objective.

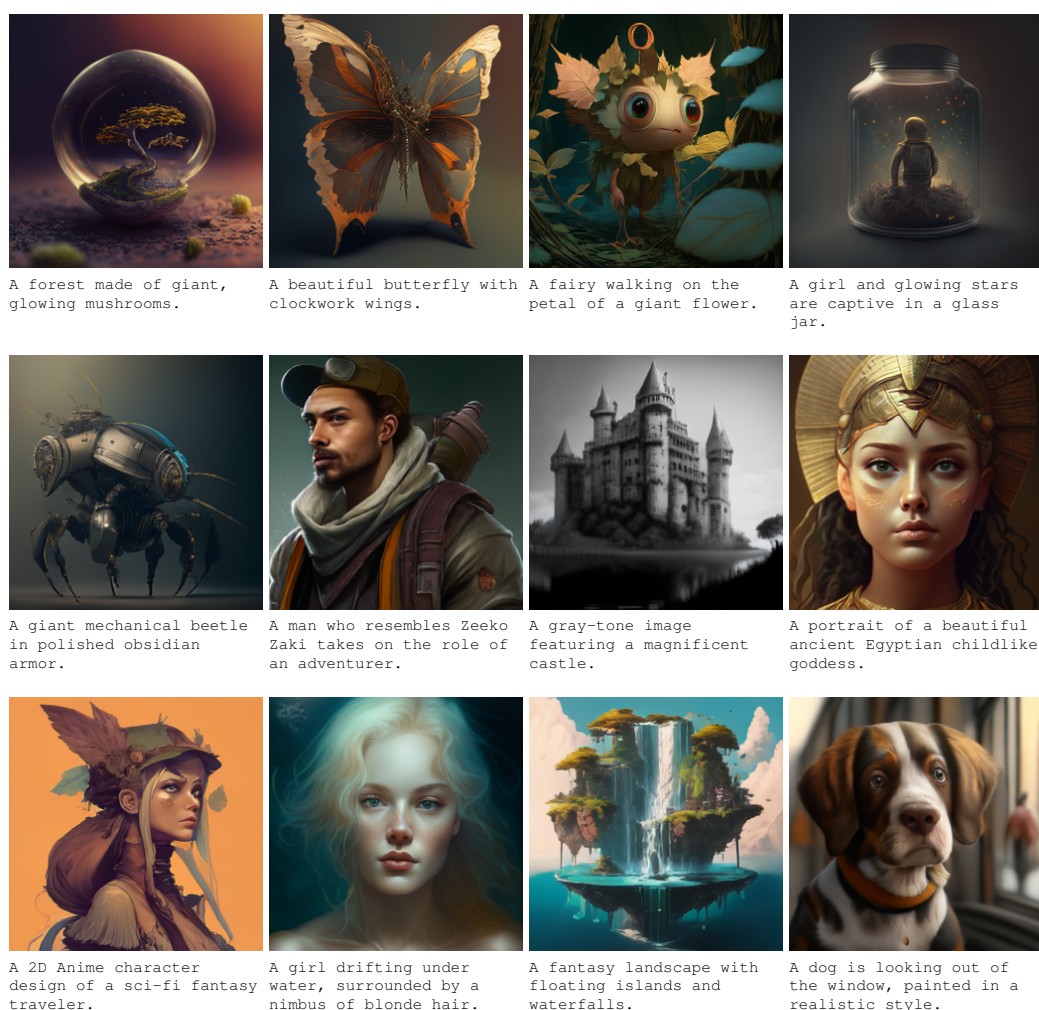

Figure 12: **Text-to-image generation.** Samples generated by ARPG-XL with 64 sampling steps. ARPG can achieve high-quality and semantically well-aligned images based on a given prompt through training on a small amount of data.

## D  GENERATION SAMPLES

To further showcase the versatility of ARPG, more examples of text-to-image generation, class-conditional image generation, controllable image generation, and zero-shot inference are provided.

**Text-to-Image Generation.**    Additional visual results for text-to-image generation, including the prompts used, are presented in Fig. 12. All images were sampled with the following parameters: 64 steps, a CFG scale of 8.0, top-k of 900, with temperature and top-p both at 1.0. These examples illustrate that ARPG is capable of producing high-quality images in various styles with a training dataset of only 4M samples. In our future work, we will further investigate the effectiveness of ARPG with a larger number of parameters and an increased volume of training data.

**Class-Conditional Generation.**    We present uncurated examples of class-conditional image generation in Fig. 13, using the same sampling parameters as set in Tab. 6. ARPG excels in both quantitative metrics and visual quality. Additionally, examples of controllable generation and zero-shot inference based on the class-conditional model are also presented in Fig. 13.

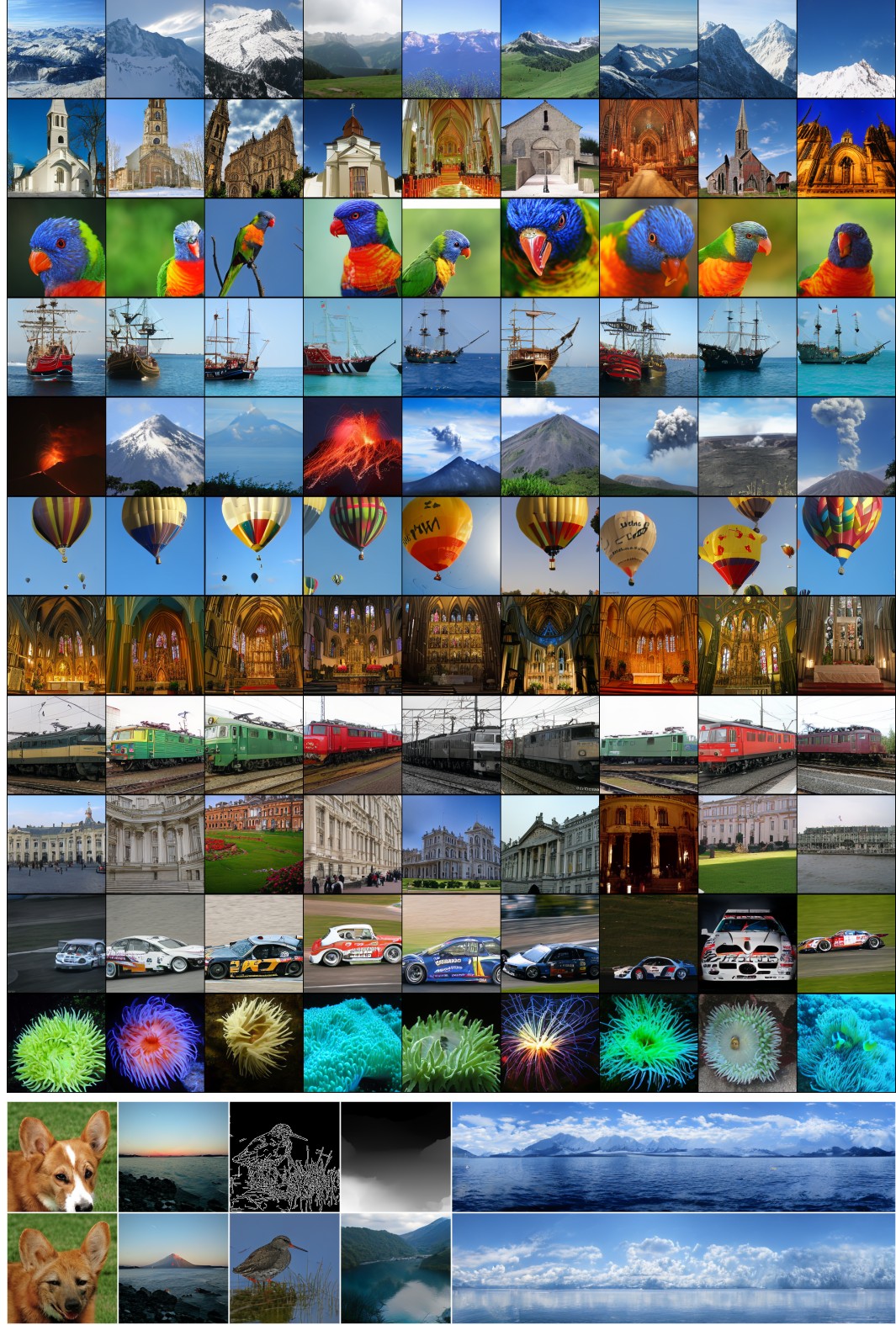

Figure 13: **Uncurated Samples in different generation tasks produced by ARPG.**

# E PSEUDO-CODE

We provide PyTorch-style pseudo code for the training and parallel decoding of our model. The classifier-free guidance is omitted for clarity.

```python
def forward_1st_decoder(model, input_ids, freqs_cis):
    x = model.embed(input_ids)
    x = model.decoder_1(x, freqs_cis)
    x = model.kv_norm(x)
    k, v = model.kv_proj(x).chunk(2, dim=-1)
    k = rearrange(k, "b t (h d) -> b t h d", h=model.num_heads)
    k = apply_rotary_emb(k, freqs_cis).transpose(1, 2)

    if model.caching: k, v = model.update_kv_cache(k, v)
    return k, v

def forward_2nd_decoder(model, k, v, freqs_cis, batch_size, num_query):
    q = model.embed(torch.full((batch_size, num_query), MASK_TOKEN_ID))
    q = model.decoder_2(q, k, v, freqs_cis)
    return model.head(model.norm(q))

def forward(model, input_ids, condition):
    B, T = input_ids.shape

    # shuffle input and RoPE using the same order.
    shuffled_ids, orders = batch_seq_shuffle(input_ids)
    freqs_cis = model.freqs_cis.unsqueeze(0).repeat(B, 1, 1, 1)
    fixed_freqs_cis = freqs_cis[:, :1, ...]
    shuff_freqs_cis = batch_seq_shuffle(freqs_cis[:, 1:, ...], orders)
    freqs_cis = torch.cat([fixed_freqs_cis, shuff_freqs_cis], dim=1)

    # prepare teacher-forcing input
    x = torch.cat([condition, shuffled_ids], dim=-1)
    k, v = forward_1st_decoder(model, x[:, :-1], freqs_cis[:, :-1, ...])
    logits = forward_2nd_decoder(
        model, k, v, freqs_cis[:, 1:, ...], batch_size=B, num_query=T)

    return cross_entropy(logits, shuffled_ids.clone().detach())

def generate(model, condition, seq_len=256, num_iter=64):
    num_samples = condition.shape[0]
    freqs_cis_ = model.freqs_cis.unsqueeze(0)
    orders = torch.rand(seq_len).argsort(dim=0) + 1

    last_pos, last_range = 0, 0
    sequences = [condition]
    for step in range(num_iter):
        num_pred = sample_schedule(step, num_iter)
        next_range = orders[range(last_pos, last_pos + num_pred)]
        last_pos += num_pred

        # the classifier-free guidance is omitted for clarity
        k, v = forward_1st_decoder(
            model, sequences[-1], freqs_cis_[:, last_range, ...])

        logits = forward_2nd_decoder(
            k, v, freqs_cis_[:, next_range, ...], num_samples, num_pred)

        tokens = sample_tokens(logits)
        sequences.append(tokens)
        last_range = next_range

    sequences = torch.cat(sequences[1:], dim=-1)
    return sequences[:, orders.argsort(dim=0)]
```

## F   DISCUSSION

In future work, we plan to extend ARPG to large-scale text-to-image synthesis and unified models for understanding and generation (Xie et al., 2024; 2025). Additionally, we will explore architectures and methods that combine the strengths of AR and diffusion models (Yu et al., 2025; Chen et al., 2025; Gao & Shou, 2025), incorporating techniques such as feature caching Ma et al. (2024); Liu et al. (2025), quantization Tao et al. (2025); Li* et al. (2025) and pruning Zhu et al. (2025); Feng et al. (2024) to further enhance efficiency.

## G   THE USE OF LARGE LANGUAGE MODELS (LLMS)

Our use of LLMs was strictly limited to proofreading for grammatical and spelling errors. LLMs made no contribution to the ideation process or the writing of the content.

