# OpenReview forum: "Autoregressive Image Generation with Randomized Parallel Decoding"
_ICLR.cc/2026/Conference — ICLR 2026 Poster_

### Official Review · Reviewer_XPoH · 2025-10-31

**Soundness:** 3
**Presentation:** 3
**Contribution:** 3
**Rating:** 6
**Confidence:** 4

**Summary:**

The paper introduces ARPG, a model designed to improve the efficiency of autoregressive image generation. It separates the generation process into two stages: Pass-1, which generates content representations using causal self-attention, and Pass-2, which decodes these representations using causal cross-attention. This separation allows for parallelization and significant improvements in inference speed and memory efficiency—up to 30× faster and 75% less memory. ARPG also maintains competitive image quality, with minimal loss of fidelity. The model excels in both speed and quality, making it suitable for real-time image generation tasks.

**Strengths:**

1. The motivation and observations in the paper are well-founded, highlighting the need for more efficient autoregressive image generation models.
2. The proposed method effectively decouples the generation process into two stages (Pass-1 and Pass-2) to improve efficiency without compromising quality.
3. The performance of ARPG is impressive, offering significant speedup and memory reduction while maintaining competitive image quality.

**Weaknesses:**

1. The description of Figure 3 is unclear, as it doesn't explicitly mention that it shows multiple attention heads, which may lead readers to think they are the same.

2. In Table 2, the citation for **NAR** appears to be incorrect. It should be updated to reference the correct source, as the current citation does not align with the intended reference or context.

3. Due to ARPG separating Pass-1 and Pass-2 into two distinct stages, each handling different tasks, the model's interpretability is limited, making it difficult to pinpoint performance issues during debugging.

**Questions:**

See weaknesses.

---

> ### Author Response · Authors · 2025-11-19
> **Response to Reviewer XPoH**
>
> We appreciate your positive assessment and constructive feedback. We have provided detailed responses to your questions below.
>
> ---
>
> **Q1: Clarity of Figure 3**
>
> Thanks for your suggestion. In the revised version of the paper, we will explicitly state in the caption that the visualization depicts several distinct heads to make this clear to the reader.
>
> ---
>
> **Q2: Incorrect Citation in Table 2**
>
> We are very grateful for your keen observation! The citation for NAR is a typo on our part. We have located the correct reference and will ensure it is fixed in the revised version of the paper.
>
> ---
>
> **Q3: Interpretability and Debugging**
>
> This is a very interesting point of discussion. We certainly see your perspective that a two-stage model could, in theory, be more complex. However, we'd like to offer a counter-perspective based on our own development experience.
>
> We actually found that this separation of concerns made the model *more* interpretable, not less.
>
> - The design forces Pass-1 to focus *only* on building a high-quality, causal content representation.
> - It forces Pass-2 to focus *only* on using that representation to make positional predictions.
>
> As we explored in our ablation studies (Tab.5), the pass-2 decoder can still achieve acceptable-quality generation relying solely on the image token embeddings, even in the complete absence of the pass-1 decoder. In our view, this clean break was actually a benefit for debugging, rather than a hindrance.
>
> ---
>
> We are very grateful for your feedback, which consists of clear and helpful clarifications that will undoubtedly strengthen the final paper. Thank you again for your time and positive assessment.

---

> > ### Comment · Reviewer_XPoH · 2025-11-27
> >
> > Thanks for the discussion. I have no more questions and will maintain my score.

---

> > > ### Author Response · Authors · 2025-11-27
> > >
> > > Thank you for your positive feedback. Your comments have been invaluable in improving the clarity and quality of our paper.

---

### Official Review · Reviewer_5zjF · 2025-10-31

**Soundness:** 4
**Presentation:** 3
**Contribution:** 4
**Rating:** 8
**Confidence:** 5

**Summary:**

This paper proposes ARPG, a novel autoregressive image generation framework that enables random-order, parallelizable token generation via a two-pass decoupled decoding mechanism: (1) a content refinement pass builds a causal, label-leakage-free memory from randomly ordered tokens; (2) a position-guided prediction pass uses [MASK] queries to decode multiple tokens in parallel. This design allows training under full AR causality while supporting flexible inference orders and block-parallel decoding. ARPG achieves good performance on ImageNet with significantly improved throughput and reduced memory, and demonstrates compelling zero-shot generalization and controllable generation.

**Strengths:**

1.Decoupling content modeling from position prediction is a conceptual advance that breaks the raster-scan bottleneck in AR vision models.
2.This paper has sufficient experimental verification and has verified the effectiveness of the method on various generation tasks such as c2i, t2i, and controllable generation.
3. This paper makes a detailed comparison of the existing AR-based models, which is helpful for understanding the uniqueness of the method proposed in this paper.

**Weaknesses:**

1. This paper contains ambiguous statements about efficiency improvement. For example, in the last sentence of the abstract section, the author claims: "achieving over a 30× speedup ... compared to representative recent autoregressive models ... ". In fact, the 30× speedup here is relative to the naive autoregressive model LlamaGen from a year ago. Compared to recent autoregressive models such as RandAR, the real advantage of ARPG in throughput is ~3× speedup. I think the author should have a more rigorous expression in the paper, especially in the explanation and comparison of quantitative indicators.

**Questions:**

I don't think this article has any fatal flaws. I only have a few minor questions to raise.
1. The resolution in the T2I generation does not seem to be mentioned in the paper.
2. I have some questions about the throughput in Table 3. Firstly, the throughput of the c2i model of LlamaGen-XL in Table 2 is 2.46, but it is 4.32 in Table 3. Why is the T2I model faster instead? I think this is abnormal. Moreover, the throughput of SD1.5 is too low. In fact, the inference efficiency of SD1.5 is higher than that of LlamaGen.

---

> ### Author Response · Authors · 2025-11-19
> **Response to Reviewer 5zjF**
>
> Thank you so much for your valuable suggestions. We are truly grateful for positive assessment of our work's novelty and contribution. We have provided detailed responses to your questions below.
>
> ---
>
> **W1: Ambiguous Efficiency Claims**
>
> We appreciate your fair and accurate criticism regarding our abstract's wording. Our intention was to highlight the dramatic leap from the standard AR paradigm, represented by LlamaGen, to our parallel approach.
>
> However, we agree with you that in the context of recent work (e.g., RandAR, NAR), the more rigorous and important comparison is against these contemporary parallel methods. The 2~3× speedup over NAR and RandAR is the more relevant SOTA comparison.
>
> We will revise the abstract and introduction in the final version to be much more precise. We will explicitly distinguish the comparison against standard AR models from the comparison against parallel AR models.
>
> ---
>
> **Q1: T2I Evaluation**
>
> 1. **T2I Resolution:** Thank you for pointing out our oversight. All text-to-image experiments were conducted at a 512x512 resolution. We will clarify this in the relevant section.
> 2. **Throughput in Table 3:** We are very grateful for your keen observation. We have investigated this and confirmed that this was indeed a transcription typo. The throughput data for Llamagen and SD1.5 in Table 3 were inadvertently swapped. We sincerely apologize for this oversight and any confusion it may have caused. This has been corrected in the revised version.
>
> ---
>
> Thank you again for your invaluable feedback. Your detailed and supportive review has been a significant help in improving the clarity and rigor of our paper.

---

> > ### Comment · Reviewer_5zjF · 2025-11-26
> > **Response to author**
> >
> > Thanks for your response. I have no other concerns about this paper. I will keep my score. I recommend that this paper be accepted

---

> > > ### Author Response · Authors · 2025-11-26
> > >
> > > Thank you very much for your recognition of our work. Your comments have been invaluable in improving the clarity and quality of our paper.

---

### Official Review · Reviewer_ePaj · 2025-10-31

**Soundness:** 4
**Presentation:** 3
**Contribution:** 3
**Rating:** 6
**Confidence:** 3

**Summary:**

This paper proposes ARPG, an efficient architecture for Random AR Image generation. First, the paper found that  (i) Rand AR require explicit positional information for the token to be predicted, and (ii) using position information tokens at the input token level hinders training efficiency and fails to achieve meaningful attention. Based on this, this paper proposes an efficient architecture for RandAR, which is built with a Pass-1 decoder trained only with visual tokens, and a Pass-2 decoder trained with cross-attention with position tokens. Specifically, the Pass-1 decoder outputs the KV cache of the current image token sequences, and the Pass-2 decoder uses this KV cache with position token queries for decoding. Experimental results showed that ARPG outperforms existing AR, PAR, and RandAR frameworks in terms of speed and quality across several benchmarks.

**Strengths:**

- The paper is well-written and easy to understand. The introduction, observations, and methodology are intuitive and clear.
- The paper presents interesting insights and observations regarding the characteristics of RandAR. To the best of my knowledge, these findings are novel.
- The proposed method is intuitively derived from the observations and simple.
- Experimental results show that ARPG outperforms existing AR models across multiple benchmarks.

**Weaknesses:**

- **Integrability with LLMs** : A key advantage of (raster) image AR or RandAR, which rely solely on self-attention decoder architecture, is that they can be easily integrated with existing LLM architectures without any changes. However, ARPG's reliance on cross-attention requires LLM architecture modification or, at least, fine-tuning by attaching Pass-2 decoder in parallel to the existing models.

- **T2I evaluation** : Experimental results on T2I evaluation are limited. In Tab.3, the model was only trained on a relatively small number of samples (4M) and actually shows worse performance. It is difficult to evaluate whether this architecture can be extended to large-scale T2I.

**Questions:**

- What are the advantages of AR modeling of image tokens over multi-modal discrete diffusion models like [1]? They also support the parallel generation of visual tokens. While I am aware that AR has advantages in KV-cache efficiency, I am just curious about the author's perspective.

- In Fig. 9 (b) and (c), what specific reason makes fewer generation steps actually improve the generation quality? Doesn't this result indicate that the current ARPG training is unstable?

[1] Yu, Runpeng, Xinyin Ma, and Xinchao Wang. "Dimple: Discrete diffusion multimodal large language model with parallel decoding." arXiv preprint arXiv:2505.16990 (2025).

---

> ### Author Response · Authors · 2025-11-19
> **[1/2] Response to Reviewer ePaj**
>
> Thank you so much for your insightful review. We are truly grateful for positive assessment of our work's novelty and contribution. We have provided detailed responses to your questions below.
>
> ---
>
> **W1: Integrability with LLMs**
>
> That's a very insightful point about the elegance of pure self-attention and its seamless fit with existing LLMs. We appreciate this concern.
>
> We would argue that ARPG *retains this key advantage* and is, in spirit, still a "decoder-only" philosophy. While it uses a pass-2 decoder with cross-attention, the *entire pipeline* remains causal and autoregressive. Both Pass-1 and Pass-2 are decoders.
>
> We believe this architecture is actually a very natural fit for LLM integration. Unlike models that fuse transformer encoders with decoders (e.g., diffusion), our approach simply adds a second, very lightweight (only have query and output projection in an attention layer) decoding pass . This design doesn't fundamentally break the paradigm and would be, in our view, straightforward to attach to an existing multimodal LLM architecture. It avoids the need for a completely different model type (like an encoder) and all the complexities that come with that joint training.
>
> ---
>
> **W2: T2I Evaluation**
>
> We agree that our T2I experiments in Table 3 are preliminary and trained on a limited dataset. Our primary goal with this experiment was not to build a SOTA T2I model (which is a massive undertaking), but rather to demonstrate the *flexibility and extensibility* of the ARPG framework. We wanted to show that our architecture *can* handle text conditioning and is not limited to class-conditional generation.
>
> We hope this clarifies our original motivation for exploring T2I. We acknowledge that we were temporarily unable to perform large-scale scaling, a limitation imposed by our available computational resources. We kindly ask for your understanding on this point. In future work, we are committed to further exploring ARPG's capabilities for both large-scale T2I and as a unified model for understanding and generation.

---

> ### Author Response · Authors · 2025-11-19
> **[2/2] Response to Reviewer ePaj**
>
> **Q1: Advantages of AR vs. Discrete Diffusion**
>
> We believe the decoder-only, next-token prediction paradigm is appealing because its training is simpler, more direct, and highly effective. It doesn't rely on hand-crafted priors (like specific mask ratios or attention masks), which, in our view, makes it a more natural fit for extension into multimodal models and unified understanding and generation architectures.
>
> Of course, the point you mentioned about native support for the KV cache is one we also see as a massive advantage. While many attempts to accelerate diffusion or MAR try to retrofit caching solutions, AR models support it natively.
>
> Furthermore, the acceleration of causal attention is far more flexible (e.g., KV cache pruning, quantization, and compression) than that of bidirectional attention. This computational efficiency also allows AR models to readily support speculative decoding. This is a benefit that the compute-intensive nature of full-attention diffusion models makes difficult to leverage.
>
> We regard AR and diffusion not as mutually exclusive, but as methodologies with potential for effective synergy. We will add citations and discussion on this topic in the revision, and we look forward to future developments combining these paradigms.
>
>
> ---
>
> **Q2: Fewer Steps Improving Quality**
>
> We appreciate your insightful question. We wish to clarify that this phenomenon stems from the inherent characteristics of our parallel decoding method rather than training instability. We have provided a comprehensive analysis with data visualization in the **revised Appendix (Page 19)**. Here is a summary of our findings:
>
> - **From a reconstruction perspective**: Our reconstruction analysis (Fig. 14) reveals that step count has a negligible impact on reconstruction quality once a high-quality prefix is established. This identifies the tokens sampled in the *early stages* as the primary determinant of final generation quality.
> - **From an entropy perspective**: We analyzed the entropy of the model’s predicted logits, as shown in Fig. 15 and Fig. 16. In the early stages of generation, the model has access to very limited context. We found that the logits distribution at this point is *extremely uniform* (Fig. 16), meaning early token generation is akin to *random sampling*. However, as more tokens are decoded, a clear divergence appears:
>     - The setting with fewer steps (e.g., 32 steps) rapidly develops logits spikes (low entropy), indicating the model is becoming highly confident about specific tokens, thus yielding a relatively deterministic prefix rather than one from random sampling.
>     - In contrast, the setting with more steps maintains a more uniform logits distribution (high entropy), which remains closer to random sampling, resulting in poor quality for the initially generated tokens.
>
> This result reflects a trade-off in the sampling strategy rather than instability. If the number of steps is too low (e.g., 16 steps), the model may become *overconfident*, exhibiting high confidence even for incorrectly predicted tokens, which in turn also aggravates error accumulation.
>
> ---
>
> We hope this clarifies your concerns and provides context for our design choices. We are very grateful for your time and the thoughtful, high-level questions, which have helped us think more deeply about the position of our work.

---

> > ### Comment · Reviewer_ePaj · 2025-11-26
> > **response to author**
> >
> > I appreciate the authors' thoughtful response. While I still have minor concerns regarding W1, my other concerns have been resolved. Overall, I think this is a good paper and above the borderline. I will maintain my positive score.

---

> > > ### Author Response · Authors · 2025-11-26
> > >
> > > We sincerely appreciate your positive evaluation and support. Regarding the perspective mentioned in W1, we recognize its value and will incorporate a discussion about this viewpoint in the final version. Thank you again for your constructive feedback.

---

### Official Review · Reviewer_SipR · 2025-11-03

**Soundness:** 3
**Presentation:** 3
**Contribution:** 2
**Rating:** 6
**Confidence:** 3

**Summary:**

This work introduces ARPG (Autoregressive Randomized Parallel Generation), a method that incorporates a decoder-decoder architecture design and decouples positional guidance from content representation.

**Strengths:**

1. The motivation is clear, where the input sequence redundancy is a bottleneck for RandAR.

2. The experiments conducted on class-to-image generation, controllable generation, zero-shot generalization, and text-to-image generation verify the effectiveness of ARPG.

**Weaknesses:**

1. At the same level of model size, e.g. 320M in Table 2, ARPG outperforms RandAR significantly in both efficiency (throughput and memory consumption) and output quality. What is the underlying reason for such improvement? Does most of the improvement come from YOCO architecture design?

2. RandAR is trained with a batch size of 1024 for 300 epochs, but ARPG is trained for 400 epochs with different batch sizes. I am wondering whether ARPG will keep the same advantage under the same training setting.

3. As a minor concern, I am curious about how ARPG with fewer sampling steps (32 steps) outperforms ARPG with more steps (64 steps)？For 256x256 image reconstruction, LlamaGen tokenizer achieves an rFID of 2.19 at 256×256 resolution. How can the gFID of ARPG surpass this upper bound?

**Questions:**

See weaknesses

---

> ### Author Response · Authors · 2025-11-19
> **[1/2] Response to Reviewer SipR**
>
> We appreciate your positive assessment and valuable suggestions. We have provided detailed responses to your questions below.
>
> ---
>
> **W1: The source of improvements:**
>
> While the YOCO-style architecture does provide a significant efficiency boost, it is not the fundamental reason for our model's gains. In fact, the YOCO architecture itself does not improve quality and may even degrade it.
>
> In our ablation study (Table 5), the variant "ARPG (+ w/o Shared KV)" does not use the full YOCO-style architecture. Even so, it is **still** substantially more efficient and performs better than RandAR with the same steps (as shown in the table below).
>
> | Model | Parameters | Steps | Throughput | Memory |
> | --- | --- | --- | --- | --- |
> | ARPG w/o Shared KV  | 343 M | 64 | 48.02 img/s | 3.83 GB |
> | RandAR | 343 M | 64 | 34.03 img/s | 7.37 GB |
>
> Regarding quality, using a YOCO-style architecture might actually lead to a slight degradation (Table 5). However, we still opted for the YOCO-style architecture in pursuit of greater efficiency.
>
> The significant improvement over RandAR stems from our fundamental contribution: decoupling positional guidance from content representation. This approach benefits both efficiency and quality:
>
> 1. Efficiency: ARPG does not need to cache KV pairs for extra positional tokens, whereas RandAR does.
> 2. Quality: During inference, ARPG's first pass decoder can generalize to intra-step bidirectional attention. We will explain the specific reasons this feature improves performance in our response to **W3**.
>
> ---
>
> **W2: The fairness of comparison:**
>
> This is a very fair point, and we appreciate you flagging this potential discrepancy. Our choice of 400 epochs was to follow the setting of recent work (e.g., RAR). We acknowledge that the field has been somewhat inconsistent on this, with training schedules varying widely.
>
> To directly address your concern about fairness, we have gone back and evaluated the 300-epoch checkpoint of our model with 64 steps.
>
> | Model | Epoch 300 | Epoch 400 |
> | --- | --- | --- |
> | ARPG-L | 2.41 | 2.37 |
> | ARPG-XL | 2.01 | 1.99 |
> | ARPG-XXL | 1.89 | 1.86 |
>
> The additional 100 epochs provide a negligible improvement for this model size. The significant advantage of ARPG over RandAR is already fully established at 300 epochs.

---

> ### Author Response · Authors · 2025-11-19
> **[2/2] Response to Reviewer SipR**
>
> **W3: The 32 vs. 64 steps and gFID vs. rFID:**
>
> 1. **32 vs. 64 steps:** We appreciate your insightful question. We have provided a comprehensive analysis with data visualization in the **revised Appendix (Page 19)**. Here is a summary of our findings:
>
> - **From a reconstruction perspective**: Our reconstruction analysis (Fig. 14) reveals that step count has a negligible impact on reconstruction quality once a high-quality prefix is established. This identifies the tokens sampled in the *early stages* as the primary determinant of final generation quality.
> - **From an entropy perspective**: We analyzed the entropy of the model’s predicted logits, as shown in Fig. 15 and Fig. 16. In the early stages of generation, the model has access to very limited context. We found that the logits distribution at this point is *extremely uniform* (Fig. 16), meaning early token generation is akin to *random sampling*. However, as more tokens are decoded, a clear divergence appears:
>     - The setting with fewer steps (e.g., 32 steps) rapidly develops logits spikes (low entropy), indicating the model is becoming highly confident about specific tokens, thus yielding a relatively deterministic prefix rather than one from random sampling.
>     - In contrast, the setting with more steps maintains a more uniform logits distribution (high entropy), which remains closer to random sampling, resulting in poor quality for the initially generated tokens.
>
> Nevertheless, reveals a critical trade-off: if the number of steps is too low (e.g., 16 steps), the model may become overconfident, exhibiting high confidence even for incorrectly predicted tokens, which in turn also aggravates error accumulation.
>
> ---
>
> 2. **gFID vs. rFID:** Thank you for your observation on this point. It is indeed a phenomenon that can easily cause confusion (in fact, we had the same doubts ourselves during the early stages of our research). It is important to note that rFID and gFID are derived from different computational methods.
>
> - The rFID metric is typically calculated using the ImageNet validation set as a reference.
> - In contrast, gFID calculation generally adheres to the ADM[1] framework, which uses virtual ImageNet statistics. Consequently, these two metrics are **not directly comparable**.
>
> In fact, the observation that gFID < rFID is quite common in related works. For example, RandAR-XXL's gFID (2.15) is lower than LlamaGen's rFID (2.19), and RAR's gFID (1.48) is significantly lower than its own rFID (2.28).
>
> [1] Prafulla Dhariwal, Alex Nichol, Diffusion Models Beat GANs on Image Synthesis, arXiv:2105.05233.
>
> ---
>
> We hope these clarifications fully address your concerns and help demonstrate the soundness and contribution of our work. We are very grateful for your time and the insightful feedback, which has certainly helped us strengthen the paper.

---

### Author Response · Authors · 2025-12-01
**Summary of Review Status**

We sincerely appreciate the time and efforts of the AC and all reviewers. To assist the AC and future readers, we provide a brief summary of the current review status:

We initially received positive ratings from all reviewers (Scores: 8/6/6/6).

- Reviewers SipR, ePaj, and XPoH commended the work as well-motivated with clear observations.
- All reviewers acknowledged the effectiveness of our method and the extensiveness of the experiments.
- Reviewers ePaj and 5zjF highlighted the novelty and uniqueness of our approach.

&nbsp;

Regarding the shared question raised by ePaj and SipR on why fewer steps (32) outperform more steps (64), we have added a comprehensive two-page analysis in the Appendix (Pages 19-20) to investigate this phenomenon from multiple perspectives.

&nbsp;

We summarize the key points from our discussion with the reviewers:

- **Reviewer 5zjF:** We clarified the efficiency claims in the Abstract, supplemented T2I experiment details, and corrected the typo in Table 3. The reviewer confirmed all concerns were resolved and *explicitly recommended our work*.
- **Reviewer ePaj:** Beyond the common question, we clarified that our T2I experiments aim to verify *flexibility and extensibility* rather than chase SOTA results. Regarding W1 (Integrability with LLMs), while the reviewer retains a minor reservation, the reviewer explicitly *stated the paper is "good"* with all other concerns resolved. **Note:** We respectfully suggest W1 is an open discussion for future work, as it falls outside the primary scope of this work, which focuses on visual AR model efficiency.
- **Reviewer XPoH:** We clarified the caption of Fig. 3, corrected a citation typo in Table 2, and elaborated on interpretability. The reviewer confirmed all concerns were resolved.
- **Reviewer SipR:** We provided clarifications on the core reasons for performance gains and experimental fairness (along with extra experiments). Regrettably, the reviewer could not participate in the discussion due to system closure. The explanation for the question regarding step counts has already been accepted by Reviewer ePaj.

We once again thank the AC and reviewers for their consideration.

---

### Meta-Review · Area_Chair_gB5Y · 2026-01-07

**Summary:**

The paper presents a novel and effective autoregressive model for image generation, and while several minor concerns were raised, they were largely addressed in the rebuttal. The clarifications on efficiency claims, model training fairness, and step count effects helped mitigate some of the key issues. As such, the paper is recommended for acceptance, with the expectation that the authors will make minor revisions based on reviewer feedback, particularly around improving clarity and refining the claims in the abstract and comparison sections.

**Reviewer Concerns:**

Reviewer SipR’s and ePaj's questions about efficiency, training fairness, and fewer steps improving quality were well explained, while 5zjF’s concerns about throughput and efficiency claims were corrected. Reviewer XPoH's issues with figure clarity and model interpretability were also resolved. Given these improvements, no major concerns remain.

**Reviewer Scores:**

Based on the authors' responses, all reviewers are likely to maintain their current scores.

---

### Decision · Program_Chairs · 2026-01-26

Accept (Poster)